# *SLC25A12* Missense Variant in Nova Scotia Duck Tolling Retrievers Affected by Cerebellar Degeneration—Myositis Complex (CDMC)

**DOI:** 10.3390/genes13071223

**Published:** 2022-07-09

**Authors:** Matthias Christen, Stefan Rupp, Iris Van Soens, Sofie F. M. Bhatti, Kaspar Matiasek, Thilo von Klopmann, Vidhya Jagannathan, Indiana Madden, Kevin Batcher, Danika Bannasch, Tosso Leeb

**Affiliations:** 1Institute of Genetics, Vetsuisse Faculty, University of Bern, 3001 Bern, Switzerland; matthias.christen@vetsuisse.unibe.ch (M.C.); vidhya.jagannathan@vetsuisse.unibe.ch (V.J.); 2Neurology Department, Tierklinik Hofheim GbR, 65719 Hofheim am Taunus, Germany; s.rupp@tierklinik-hofheim.de (S.R.); t.vonklopmann@tierklinik-hofheim.de (T.v.K.); 3Companion Animal Internal Medicine Section, Faculty of Veterinary Medicine, Liège University, 4000 Liège, Belgium; iris.vansoens@uliege.be; 4Dierenziekenhuis Hart van Brabant, 5144 AM Waalwijk, The Netherlands; 5Small Animal Department, Faculty of Veterinary Medicine, Ghent University, 9820 Merelbeke, Belgium; sofie.bhatti@ugent.be; 6Section of Clinical and Comparative Neuropathology, Centre for Clinical Veterinary Medicine, Ludwig-Maximilians Universität München, 80539 Munich, Germany; kaspar.matiasek@neuropathologie.de; 7Department of Population Health and Reproduction, University of California-Davis, Davis, CA 95616, USA; iemadden@ucdavis.edu (I.M.); klbatcher@ucdavis.edu (K.B.); dlbannasch@ucdavis.edu (D.B.)

**Keywords:** *Canis lupus familiaris*, neurology, seizure, N-acetyl aspartic acid, aralar, precision medicine, animal model

## Abstract

We investigated two litters of distantly related Nova Scotia Duck Tolling Retrievers (NSDTR), of which four puppies were affected by cerebellar signs with or without neuromuscular weakness. The phenotype was termed cerebellar degeneration—myositis complex (CDMC). We suspected a heritable condition and initiated a genetic analysis. The genome of one affected dog was sequenced and compared to 565 control genomes. This search yielded a private protein-changing *SLC25A12* variant in the affected dog, XM_038584842.1:c.1337C>T, predicted to result in the amino acid change XP_038440770.1:(p.Pro446Leu). The genotypes at the variant co-segregated with the phenotype as expected for a monogenic autosomal recessive mode of inheritance in both litters. Genotyping of 533 additional NSDTR revealed variant allele frequencies of 3.6% and 1.3% in a European and a North American cohort, respectively. The available clinical and biochemical data, together with current knowledge about *SLC25A12* variants and their functional impact in humans, mice, and dogs, suggest the p.Pro446Leu variant is a candidate causative defect for the observed phenotype in the affected dogs.

## 1. Introduction

In dogs, advances in genetic research have led to the discovery of diverse genetic variants causing neurologic phenotypes [1]. Many of these inherited neurologic diseases are restricted to one of approximately 300 recognized dog breeds. In the Nova Scotia Duck Tolling Retriever (NSDTR) breed, a degenerative encephalopathy with presumed autosomal recessive inheritance has previously been described (OMIA 002055-9615) [2]. Affected dogs show episodes of marked movements during sleep, increased anxiety, noise phobia, and gait abnormalities. MRI and post mortem examination showed symmetrical changes in the cerebellar caudate nuclei of affected dogs [2]. The causal genetic variant for the described disease has not been reported so far.

The current investigation was initiated after the presentation of an NSDTR litter, in which a single puppy showed a combination of cerebellar and neuromuscular signs that were not consistent with the previously published degenerative encephalopathy in the NSDTR [2]. Hence, a new inherited disease was suspected. During our study, a second, distantly related litter with three additional affected dogs was identified and included in the analysis. The goal of this project was to grossly characterize the clinical phenotype and to investigate a possible genetic cause.

## 2. Materials and Methods

### 2.1. Clinical and Pathological Examination

A clinical and neurological examination, complete blood count, and serum biochemistry was performed in all 4 affected dogs. Electromyography (EMG), MRI of the head, cervical cerebrospinal fluid (CSF) exam, PCR analyses on infectious diseases, and muscle and nerve biopsies were performed in one dog with cerebellar signs combined with neuromuscular weakness. One dog was euthanized, and a full necropsy was performed, including investigation of skeletal muscles and the central and peripheral nervous system.

### 2.2. Animal Selection for Genetic Analysis

The study was conducted with a total of 563 NSDTR samples. Thirty dogs belonged to two extended families of European origin with a total of four affected puppies. The remaining 533 dogs originated from sample donations to the Vetsuisse biobank (380 dogs of European origin) and the UC Davis Bannasch biobank (153 dogs of mostly North American origin). These 533 dogs were designated as population controls without reports of a similar specific neurologic disease.

### 2.3. DNA Extraction

Genomic DNA was extracted from EDTA blood samples according to standard methods using the Maxwell RSX Whole Blood DNA kit in combination with the Maxwell RSC instrument (Promega, Dübendorf, Switzerland) or using the Gentra Puregene DNA purification extraction kit (Qiagen, Valencia, CA, USA).

### 2.4. Whole Genome Sequencing

An Illumina TruSeq PCR-free DNA library with ~420 bp insert size was prepared from one affected dog. We collected 260 million 2 × 150 bp paired end reads on a NovaSeq 6000 instrument (29.1× coverage). Mapping and alignment to the UU_Cfam_GSD_1.0 reference genome assembly were performed as described [3]. The sequence data were deposited under the study accession PRJEB16012 and the sample accession SAMEA10644737 at the European Nucleotide Archive.

### 2.5. Variant Calling

Variant calling was performed using GATK HaplotypeCaller [4] in gVCF mode as described [3]. To predict the functional effects of the called variants, SnpEff [5] software together with NCBI annotation release 106 for the UU_Cfam_GSD_1.0 genome reference assembly was used. For variant filtering, we used 565 control genomes from dogs of different breeds (Appendix A).

### 2.6. Gene Analysis

We used the UU_Cfam_GSD_1.0 dog reference genome assembly and NCBI annotation release 106. Numbering within the canine *SLC25A12* gene corresponds to the NCBI RefSeq accession numbers XM_038584842.1 (mRNA) and XP_038440770.1 (protein).

### 2.7. Database Searches and In Silico Functional Predictions

The Genome Aggregation Database (gnomAD) [6] and Online Mendelian Inheritance in Man and Animals databases [7,8] were searched for corresponding variants in the human and domestic animal *SLC25A12* genes. Additionally, the Mouse Genome Informatics Web Site was screened for corresponding phenotypes in mice [9]. PredictSNP [10], PROVEAN [11], and MutPred2 [12] were used to predict biological consequences of the discovered candidate protein variant. The human and canine SLC25A12 proteins both comprise 678 amino acids, of which 663 (98%) are identical between dogs and humans.

### 2.8. PCR and Sanger Sequencing

Primers 5′-TCA TCC CTG TGA GCT CCT CT-3′ (Primer F) and 5′-GAA GCC TGG TTT CCA CAT TC-3′ (Primer R) were used for the generation of an amplicon containing the *SLC25A12*:c.1337C > T variant. PCR products were amplified from genomic DNA using AmpliTaq Gold 360 Master Mix (Thermo Fisher Scientific, Reinach, Switzerland). Direct Sanger sequencing of the PCR amplicons on an ABI 3730 DNA Analyzer (Thermo Fisher Scientific, Reinach, Switzerland) was performed after treatment with exonuclease I and alkaline phosphatase. Sanger sequences were analyzed using the Sequencher 5.1 software (Gene Codes, Ann Arbor, MI, USA).

## 3. Results

### 3.1. Clinical History and Examination

Age of onset of neurological signs was between 10 weeks and 6 months. Clinical abnormalities were restricted to the neuromuscular system in all four dogs. Neurological examination showed generalized ataxia and hypermetria, which was more pronounced in the pelvic limbs in all four dogs. Intentional head tremor was present in one dog. In two dogs, generalized neuromuscular weakness became apparent after 1 month, characterized by exercise intolerance, episodic collapse, stiff gait, and bunny hopping. Hopping was delayed in four limbs in three dogs, menace responses were absent in one dog, and decreased withdrawal reflexes were found in four limbs of three dogs.

### 3.2. Ancillary Diagnostic Investigations

Blood examination showed increases in serum creatine kinase concentrations in all four dogs (between 3 and 25 times greater than the upper reference limit). An EMG showed mild spontaneous activity in peripheral limb muscles. MRI of the brain in the same dog showed bilateral symmetrical lesions in the cerebellum and multifocal lesions in the masticatory muscles. CSF analysis and PCR analyses (Toxoplasma, Neospora, Distemper virus, Bartonella, and Tick-borne encephalitis) were all within normal limits. Muscle and nerve biopsies showed a fiber-invasive lymphohistiocytic myositis without evidence of intracellular infectious agents on histology and tissue PCR. The only other abnormality seen on postmortem examination was severe cerebellar nuclear degeneration.

Based on the available clinical and diagnostic findings, we tentatively termed the phenotype of the four affected dogs cerebellar degeneration—myositis complex (CDMC).

### 3.3. Genetic Analysis

At the beginning of the genetic analysis, we only had access to a single affected puppy and its close relatives. We obtained blood samples from the affected puppy, five unaffected littermates, and both parents. Pedigree analysis revealed a common founder of both parents and was thus suggestive of a monogenic autosomal recessive mode of inheritance of the investigated trait (Appendix A).

The genome of the affected dog was sequenced, and we searched for private homozygous variants that were not present in the genome sequences of 565 control dogs of diverse breeds (Table 1 and Appendix A).

We prioritized the resulting variants according to functional knowledge of the altered genes in the OMIM database. This process revealed a single homozygous candidate variant for the observed neurologic/neuromuscular phenotype, located in *SLC25A12*, coding for the solute carrier family 25-member 12 protein. The detected missense variant, XM_038584842.1:c.1337C>T, is predicted to result in an amino acid substitution in the protein, XP_038440770.1:(p.Pro446Leu). On the genomic level, the variant can be described as Chr36:16,504,064G>A (UU_Cfam_GSD_1.0 assembly). The other five homozygous private protein-changing variants were located in genes that are not known to have specific functions in the neuromuscular system (Appendix A).

The proline to leucine substitution at position 446 of the SLC25A12 protein was predicted to be pathogenic and deleterious by several in silico prediction tools (PredictSNP probability for pathogenicity: 87%; MutPred2 score: 0.911; PROVEAN score: −9.639). The other five private homozygous missense variants were located in genes not known to cause phenotypes similar to our case in humans, mice, or domestic animals.

We confirmed the presence of the *SLC25A12* variant in a homozygous state in the affected dog via Sanger Sequencing (Figure 1).

At this point of the genetic analysis, we became aware of the second litter of NSDTRs, in which three dogs showed a phenotype comparable with the index case. We obtained DNA samples of the second family and genotyped the variant in both families. The genotypes at the variant co-segregated with the CDMC phenotype, as expected for a monogenic autosomal recessive mode of inheritance in both families. Common ancestors were identified for three of four parents of the affected litters (Appendix A).

We genotyped 533 additional NSDTR population controls that were sampled independently of our investigation into CDMC. None of these dogs carried the mutant T-allele in a homozygous state. The carrier frequencies were 7.1% in the European and 2.7% in the North American cohort. The corresponding mutant allele frequencies were 3.6% and 1.3%, respectively (Table 2).

## 4. Discussion

In this study, we provide an initial clinical characterization of a new disease with monogenic autosomal recessive inheritance in NSDTR dogs. We tentatively termed the disease cerebellar degeneration—myositis complex (CDMC). Further investigations to characterize the phenotype in more detail, including its progression over time, are ongoing.

Whole-genome sequencing of an affected dog identified a homozygous private protein-changing variant, *SLC25A12*:c.1337C>T. Genotypes at this variant co-segregated with the phenotype in two families, and no other homozygous dog with a similar phenotype was identified in whole-genome sequence data of 565 control dogs of different breeds. Additionally, two different cohorts of breed-matched control dogs were genotyped and found to be free of additional homozygous dogs. Thus, the total number of investigated controls is in excess of 1000 dogs.

The *SLC25A12* gene encodes solute carrier family 25 member 12, which has also been termed mitochondrial aspartate–glutamate carrier 1 or aralar. The SLC25A12 protein is located in the inner mitochondrial membrane [13,14]. There, it has the function of a Ca^2+^ activated aspartate–glutamate carrier and is part of the malate–aspartate shuttle, a major NADH redox unit [15].

*Slc25a12^−/−^* mice showed growth defects, generalized tremors, postnatal lethality, impaired motor coordination, and CNS dysmyelination [16]. The clinical signs in knockout mice were associated with decreased myelin lipid synthesis and significant reduction in aspartate and NAA levels in the brain. Similar to *Slc25a12^−/−^* mice, variants in the human *SLC25A12* cause the rare developmental and epileptic encephalopathy 39 (OMIM #612949) [17]. This disease is characterized by global developmental delay, seizures, hypotonia with poor motor function, and hypomyelination on brain imaging. Analogous to the process in knockout mice, the myelination defect is speculated to reflect the impaired supply of NAA to oligodendrocytes caused by SLC25A12 deficiency [18].

In dogs, an *SLC25A12*:p.Leu349Pro missense variant with experimentally verified impairment of transport activity and documented changes in the skeletal muscle metabolome leading to a proinflammatory milieu and increased oxidative stress has been reported in Dutch Shepherd dogs with inflammatory myopathy (OMIA 002294-9615) [19]. The reported inflammatory myopathy was comparable between the affected Dutch Shepherd dogs (L349P) [19] and our findings in affected NDSTRs (P446L). In the Dutch Shepherd dogs, generalized progressive weakness was reported as the main clinical finding. Most of the detailed histological and biochemical analyses in the Dutch Shepherd study were performed on muscle biopsies. A detailed clinical neurological examination was not performed [19]. It is therefore not clear whether there are true differences in the neurological phenotype between the two dog breeds that might be due to the different missense variants and/or additional breed-specific modifiers. Given that the clinical phenotype in affected NDSTRs was quite variable, with some dogs showing more pronounced muscular weakness and other dogs showing generalized ataxia and hypermetria, we think that the existing data do not yet allow conclusive definition of the genotype–phenotype correlation in dogs. This underscores the need for more comprehensive clinical and histopathological investigations of a sufficiently large number of affected dogs.

## 5. Conclusions

The clinical, radiological and pathological presentation, together with genetic findings and existing knowledge of *SLC25A12* variants in humans, dogs, and mice, suggest the *SLC25A12*:p.Pro446Leu as a candidate causative variant for CDMC in NSDTR dogs. The identification of a candidate causative variant enables genetic testing for early diagnosis and the detection of heterozygous carriers, thus preventing the further unintentional breeding of affected dogs.

## Figures and Tables

**Figure 1 genes-13-01223-f001:**
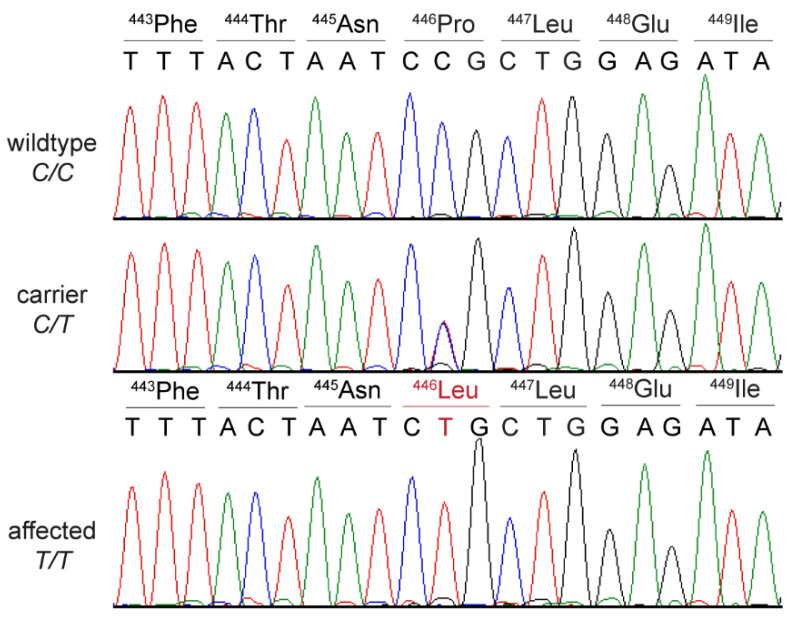
Details of the *SLC25A12*:c.1337C>T variant (p.Pro446Leu). Representative electropherograms of dogs with the three different genotypes are shown. The amino acid translations of the wild type and mutant alleles are indicated.

**Table 1 genes-13-01223-t001:** Results of variant filtering in the affected dog against 565 control genomes.

Filtering Step	Homozygous Variants	Heterozygous Variants
All variants in the affected dog	2,535,158	3,669,142
Private variants	2057	10,901
Protein-changing private variants	6	88

**Table 2 genes-13-01223-t002:** Association of the genotypes at *SLC25A12*:c.1337C>T variant with cerebellar degeneration—myositis complex (CDMC) in 563 NSDTR dogs.

Phenotype	C/C	C/T	T/T
CDMC cases (*n* = 4)	-	-	4
Non-affected family members (*n* = 26)	12	14	-
Control dogs from Europe (*n* = 380)	353	27	-
Control dogs from North America (*n* = 153)	149	4	-

## Data Availability

The accessions for the sequence data reported in this study are listed in Appendix A.

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
