# Peer review of "SLC25A12 Missense Variant in Nova Scotia Duck Tolling Retrievers Affected by Cerebellar Degeneration—Myositis Complex (CDMC)"

_genes, 2022, doi:10.3390/genes13071223_

Round 1

Reviewer 1 Report

This is a well-written and succinct manuscript describing genetic analysis of a newly described neurological disease phenotype in dogs. This reviewer has only two criticisms. In section 3.1 of results, the second sentence (lines 110-111) claims there are no clinical abnormalities, but the immediately following sentences describe a number of neurological abnormalities that are evident in the physical examination.  Please delete the sentence on lines 110-111 or say something like," clinical abnormalities were confined to the neuromuscular system".

The second criticism relates to the statement in section 3.3, lines 150-152, wherein the authors treat the 5 homozygous protein coding variants in gene other than SLC25A12 as irrelevant. Given that our knowledge of gene function and genotype/phenotype correlations is imperfect, to say the least, The readers would appreciate a list of those 5 variants and the genes affected, so that they can judge their relevance for themselves. At the very least, please give the evidence that they are not relevant to the described phenotype by showing that these variants do not segregate with the disease phenotype in the available families.

Author Response

(1)

This is a well-written and succinct manuscript describing genetic analysis of a newly described neurological disease phenotype in dogs. This reviewer has only two criticisms. In section 3.1 of results, the second sentence (lines 110-111) claims there are no clinical abnormalities, but the immediately following sentences describe a number of neurological abnormalities that are evident in the physical examination. Please delete the sentence on lines 110-111 or say something like," clinical abnormalities were confined to the neuromuscular system".

Response: Thank you very much for the compliments. We revised the conflicting statement in lines 110-111 accordingly.

(2)

The second criticism relates to the statement in section 3.3, lines 150-152, wherein the authors treat the 5 homozygous protein coding variants in gene other than SLC25A12 as irrelevant. Given that our knowledge of gene function and genotype/phenotype correlations is imperfect, to say the least, The readers would appreciate a list of those 5 variants and the genes affected, so that they can judge their relevance for themselves. At the very least, please give the evidence that they are not relevant to the described phenotype by showing that these variants do not segregate with the disease phenotype in the available families.

Response: We fully agree with the reviewer that the knowledge of gene functions and genotype/phenotype correlations is far from complete. We give a list of all private variants in Table S2, so that the interested reader can easily retrieve the other variants. (To obtain the protein-changing variants, you have to filter the column "IMPACT" for "high" and "moderate".) We added one additional sentence to the results to make this more clear.

The 5 additional homozygous private protein-changing variants were located within these genes: PGF (placental growth factor), LOC119869011_2, LOC612235, ASCL2 (achaete-scute family BHLH transcription factor 2), PKP4 (plakophilin 4). We think that PGF can be safely excluded based on its known function, while we concede that the knowledge on the other 4 genes is fragmentary.

In summary, out of the 6 genes with private homozygous variants, we have 1 that can be safely excluded (PGF), 4 that have not been connected to neuromuscular phenotypes, but for which the knowledge is very fragmentary, and 1 that has been previously linked to very similar phenotypes in humans, mice and -most importantly- also dogs. The fact that another SLC25A12 variant has been linked to a comparable disease in Dutch Shepherd dogs in our opinion justifies the prioritization of the SLC25A12 variant as the top candidate.

As we were given only 5 days to prepare the revision, we could unfortunately not experimentally genotype these variants in our samples.

Reviewer 2 Report

Very well written piece of genetic research. Research objectives and methodology are clear. Results are also comprehensive and tentative conclusions presented. My only question is below: -

Line 25: not sure what you mean by the word “private”

Author Response

Very well written piece of genetic research. Research objectives and methodology are clear. Results are also comprehensive and tentative conclusions presented. My only question is below: -

Line 25: not sure what you mean by the word “private”

Response: A private variant is restricted to a single individual or a clearly defined cohort of individuals and does not occur in a control cohort. This is a standard term in genetics that has been in use since at least 1978 (JV Neel. Rare variants, private polymorphisms, and locus heterozygosity in Amerindian populations. Am J Hum Genet 1978, 30, 465-490).